# Comparison of FLACS and BASiL Model for Ro-Pax Ferry LNG Bunkering Leak Analysis

**Boon How Lim** [1,2] and **Eddie Y. K. Ng** [1,*]

1 College of Engineering, School of Mechanical and Aerospace Engineering, Nanyang Technological University, 50 Nanyang Ave, Singapore 639798, Singapore
2 Sembcorp Marine, 80 Tuas South Boulevard, Singapore 637051, Singapore
* Correspondence: mykng@ntu.edu.sg

**Abstract:** Performing liquefied natural gas (LNG) bunkering involves the risk of accidental leakage. When released from containment, LNG rapidly vaporizes into flammable natural gas and could lead to flash fire and explosion. Hence, LNG bunkering needs to take place in an area without an ignition source called a safety zone. This study compares the safety zone estimated by the Bunkering Area Safety Information for LNG (BASiL) model with that of the computational fluid dynamic (CFD) software FLACS, for Ro-Pax ferry bunkering. Horizontal leaks covering different wind speeds in eight wind directions were compared between the two models. Additionally, a grid refinement study was performed systematically to quantify the discretization error uncertainty in the CFD. Of 24 leak cases, FLACS and the BASiL model results agreed on 18 cases. In three cases validation was inconclusive due to the CFD error uncertainty. The BASiL model underestimated the safety zone distance in three cases compared with FLACS. Future work would be to perform a higher grid refinement study to confirm inconclusive comparison and examine ways to reduce gas dispersion spread for the worst result.

**Keywords:** LNG leak; BASiL model; FLACS; Ro-Pax ferry

## 1. Introduction

Liquefied natural gas (LNG) will become an important energy source as the world aims to lower its greenhouse gas emissions [1]. According to the DNV Energy Transition Outlook 2020, the marine industry. especially shipping, will switch from predominantly oil fuel to low carbon fuels and LNG [2]. When performing LNG bunkering, ship owners have a responsibility to ensure safety when an accidental LNG leak occurs. The specification for the bunkering of liquefied natural gas-fueled vessels (ISO 20519) requires risk assessment of accidental LNG leaks, especially for flammable gas spreading [3]. LNG rapidly vaporizes into natural gas (NG) when spilled, creating a dispersing flammable cloud. This flammable cloud can lead to flash fire or explosion if it encounters an ignition source. Therefore, it is important to set up a safety zone that does not contain any ignition sources in which the LNG can disperse. However, safety zone evaluation is challenging because factors such as leak rate, leak frequency, wind speed, wind direction, and the surrounding geometry will influence the gas dispersion [4].

Walter and colleagues have summarized the two main approaches developed to study the dispersion of LNG leaks [5]. The two approaches are the integral model and the Navier–Stokes model. The integral model uses a simplified conservation equation with a mathematical complexity of one dimension. Well-known commercial software which uses the integral model includes PHAST [6] and ALOHA [7]. The Navier–Stokes model covers the complete representation of fundamental fluid dynamics over three dimensions, solving the time-dependent conservation equation of momentum, mass, energy, and species. While the Navier–Stokes model offers higher accuracy, it is time-consuming and often the

integral model is used for preliminary risk assessment. The Navier–Stokes model is used in computational fluid dynamic (CFD) software. Please see the CFD papers by Mattia and colleagues on safety zone evaluation for LNG pool leaks within a harbor [8,9]. The software PHAST and Fire Dynamics Simulator (FDS) were utilized to predict the extent of LNG dispersion during LNG leak scenarios. Three types of bunkering operation, LNG unloading operation, Shore to Ship, and Truck to Ship were considered for leak simulation. The heat flux from the ground was found to be the most impactful parameter on the size of the flammable plume size. The study also highlighted how uneven terrain aids in flammable gas mixing efficiency, thereby limiting the flammable area to the proximity of the ship.

Marko and colleagues compared the LNG pool dispersion results from PHAST, FDS, and Ansys Fluent at the port of Koper, Slovenia [10]. The results show that all three models produce comparable results for the gas dispersion distance on flat terrain. It was concluded that PHAST is not reliable for LNG leaks that occur near lower structures such as piers, houses, and moored ships. This is because the integral model does not consider the heights of geometry and will not capture the stratification of a vaporization gas cloud. The presence of the ship significantly influences the dispersion of the gas cloud shape and length. This paper concluded that CFD should be considered for gas dispersion in ports with a defined gas evaporation rate obtained from experimental tests or simulated separately and validated for a specific spill case.

The Society for Gas as Marine Fuel (SGMF) has developed an approach named Bunkering Area Safety Information for LNG (BASiL) to estimate the safety zone for bunkering activity [11]. BASiL contains a database of 1.4 million combinations of eight input parameters: location (latitude and longitude); amount of LNG transferred and the duration of the transfer; LNG supply temperature and pressure; LNG composition; transfer pressure; transfer system elevation from the ground; hose/transfer system diameter; and emergency shut down (ESD) method. The model interpolates simultaneously within the most appropriate points in the database to derive the safety zone for LNG bunkering. BASiL neither falls under the integral model nor the Navier–Stokes model category and is akin to a data analysis approach. Given the novelty of the BASiL approach to predict LNG dispersion, it will be worthwhile to validate it with CFD software. This paper compares the safety zone predicted by BASiL and the Flame Acceleration Simulator (FLACS) for a Ro-Pax ferry bunkering operation [12]. FLACS is developed by the company Gexcon and has been validated extensively with LNG spill experiments such as Burro [13], Maplin Sands [14], and Falcon [15].

## 2. Materials and Methods

The leak scenario used for comparison is the LNG bunkering of a Ro-Pax ferry in Hong Kong. The Ro-Pax ferry fueling was performed with a 4-inch hose, using a pump from two LNG road tankers filled at 2.5 bars and pumped to 6 bars. The ferry's manifold is located in the hull, 10 m below the main deck. The hose has a minimum height above the ground of 0.1 m. The emergency shut down (ESD) system is semi-automatic, acting after 120 s. Considering a 6 mm leak hole in a 4-inch hose, the BASiL model predicted a horizontal jet radius of 27 m from the leak source [11]. Figure 1 is an illustration of the bunkering setup and the semi-circle in pink represents the safety zone.

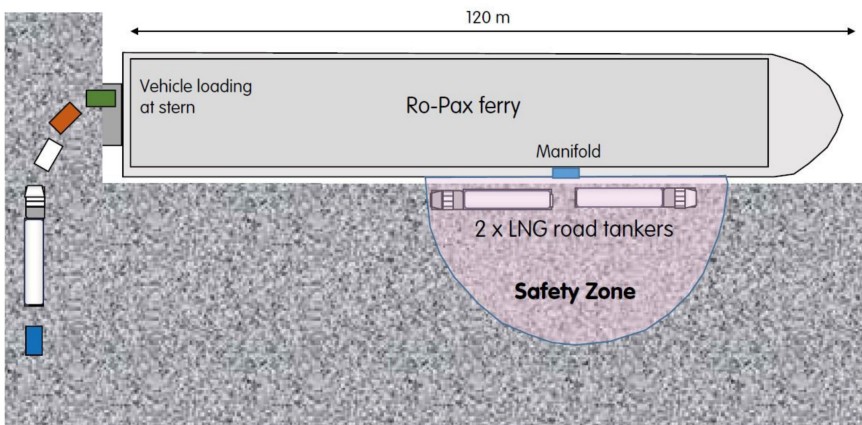

**Figure 1.** BASiL safety zone for Ro-Pax ferry LNG bunkering in Hong Kong (extracted from SGMF manual).

The CFD simulation was performed using a Dell Precision T7820, equipped with dual Intel Xeon Gold 6150 2.7 GHz processor and 128 GB (8 × 16 GB) 2666 MHz DDR4 RAM. CFD software FLACS version 20.1 simulated the leak scenario of the Ro-Pax ferry for comparison. FLACS used the finite volume method on a structured cartesian grid to solve Favre averaged transport equations for mass, momentum, enthalpy, turbulent kinetic energy, the rate of dissipation of turbulent kinetic energy, and mass-fraction of fuel. The porosity concept was used to capture the congestion effect of geometries within the cartesian grid which affect gas dispersion. The governing equation for gas transport equation is as follows:

$$\frac{\partial}{\partial t}(\beta_v \rho Y_{fuel}) + \frac{\partial}{\partial x_j}\left(\beta_j \rho u_j Y_{fuel}\right) = \frac{\partial}{\partial x_j}\left(\beta_j \frac{\mu_{eff}}{\sigma_{fuel}}\frac{\partial Y_{fuel}}{\partial x_j}\right) + R_{Fuel} \tag{1}$$

$R_{fuel}$ is the fuel reaction rate (zero since LNG is not reacting with air). $\sigma_{fuel}$ is the Prandtl/Schmidt number pre-defined as 0.7. $\beta_v$ is the volume porosity of the mesh. $\beta_j$ is the area porosity of the mesh in the j direction. $\rho$ is the air density. $Y_{fuel}$ is the gas concentration. $u_j$ is the velocity of the wind in the j direction. $x_j$ is the axis in the j direction. $\mu_{eff}$ is the effective viscosity.

The effective viscosity, $\mu_{eff}$, is as follows:

$$\mu_{eff} = \mu + \rho C_\mu \frac{k^2}{\varepsilon} \tag{2}$$

$\mu$ is the gas viscosity and $C_\mu$ is pre-defined as 0.09. k is turbulent kinetic energy and $\varepsilon$ is the dissipation rate of turbulent kinetic energy. The turbulence effect on gas dispersion is based on the k-$\varepsilon$ turbulence model.

The transport equation for the dissipation rate of turbulent kinetic energy is as follows:

$$\frac{\partial}{\partial t}(\beta_v \rho \varepsilon) + \frac{\partial}{\partial x_j}\left(\beta_j \rho u_j \varepsilon\right) = \frac{\partial}{\partial x_j}\left(\beta_j \frac{\mu_{eff}}{\sigma_\varepsilon}\frac{\partial \varepsilon}{\partial x_j}\right) + \beta_v P_\varepsilon - C_{2\varepsilon}\beta_v \rho \frac{\varepsilon^2}{k} \tag{3}$$

$C_{2\varepsilon}$ is pre-defined as 1.92 and $P_\varepsilon$ is the production of dissipation.

The transport equation for turbulent kinetic energy is as follows:

$$\frac{\partial}{\partial t}(\beta_v \rho k) + \frac{\partial}{\partial x_j}\left(\beta_j \rho u_j k\right) = \frac{\partial}{\partial x_j}\left(\beta_j \frac{\mu_{eff}}{\sigma_k}\frac{\partial k}{\partial x_j}\right) + \beta_v P_K - \beta_v \rho \varepsilon \tag{4}$$

$P_K$ is the summation of the production of turbulent kinetic energy due to shear stress and buoyancy. $\sigma_k$ is pre-defined as 1.

The SIMPLE (Semi-Implicit Method for Pressure-Linked Equation) algorithm is applied for pressure and velocity correction [16]. The FLACS numerical model employs a second-order hybrid scheme (second order upwind and second order central difference, with delimiters for certain conditions) for diffusive/convective flux equations. The first order backward Euler scheme is applied for the time-stepping scheme. The convergence criterion is mass residual and shall be less than $10^{-4}$.

For the wind boundary condition, the wind velocity at an elevation z is as follows [17]:

$$U(z) = \frac{u^*}{\kappa}\left[\ln\left(\frac{(z-z_d)+z_0}{z_0}\right) - \psi_u(z)\right] \tag{5}$$

$z_0$ is atmospheric roughness length and $z_d$ is canopy height (0 because the simulation takes place on flat ground). $z_0$ and $z_d$ are user-defined. $\kappa$ is the von Kármán constant given as 0.41.

$u^*$ is the friction velocity and is as follows:

$$u* = \frac{U_0\kappa}{\ln\left(\frac{(z_{ref}-z_d)+z_0}{z_0}\right) - \psi_u(z_{ref})} \tag{6}$$

$U_0$ is the wind speed at reference height $z_{ref}$. $U_0$ and $z_{ref}$ are user-defined.

In this paper, Pasquill class D (neutral) is applied and the function $\psi_u(z)$ is given zero value.

The turbulence parameter profiles of the wind boundary at elevation, z, are as follows:

$$k(z) = \begin{cases} 6(u*)^2 & \text{if } z \le 0.1h \\ 6(u*)^2\left(1-\frac{z}{h}\right)^{1.75} & \text{if } z > 0.1h \end{cases} \tag{7}$$

$$\varepsilon(z) = \begin{cases} \frac{(u*)^3}{\kappa z}\left(1.24+4.3\frac{z}{h}\right)^2 & \text{if } z \le 0.1h \\ \frac{(u*)^3}{\kappa z}\left(1.24+4.3\frac{z}{L}\right)\left(1-0.85\frac{z}{h}\right)^{3/2} & \text{if } z > 0.1h \end{cases} \tag{8}$$

$u^*$ is the friction velocity. $k$ is turbulent kinetic energy and $\varepsilon$ is the dissipation rate of turbulent kinetic energy. $\kappa$ is the von Kármán constant given as 0.41. h is the height of the atmospheric mixing layer [18]. L is the Monin–Obukhov length.

For inflow/parallel boundaries, Equations (5)–(8) are applied to set the boundary condition. For the outflow boundaries, the momentum and continuity equations are solved. For more information on FLACS, please refer to the FLACS-CFD 20.1 User Manual, 2020 [12].

The ship model was based on Adinda Windu Karsa, a Ro-Pax ferry working in the Indo-Java Sea [19]. The ferry measures 114.8 m by 22 m and the freeboard height is approximately 13.2 m above the sea surface. Figure 2 is a picture of the Ro-Pax ferry model used in the CFD simulation.

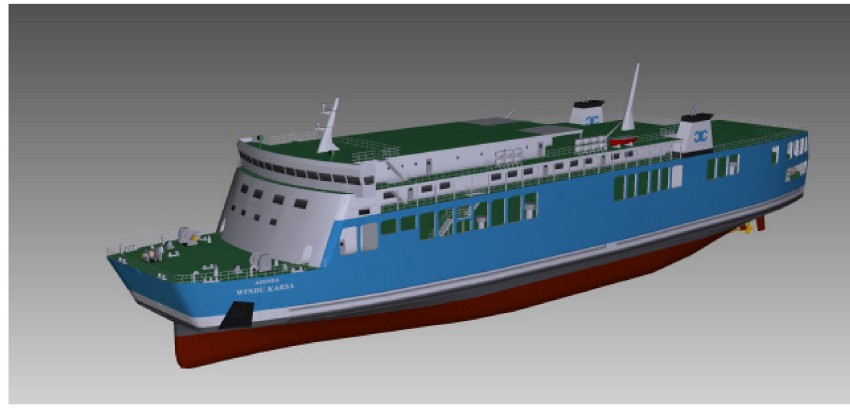

**Figure 2.** Model of Ro-Pax ferry, Adinda Windu Karsa, used in CFD simulation.

The LNG truck tankers model was based on the Mercedes Benz Actros truck series. The two trucks were modeled back-to-back, each connected to a trailer loaded with two 20-foot tanks. A pump was located between the two road tankers to transfer the LNG into the Ro-Pax ferry. The road tankers, pump, and Ro-pax vessel positioning were identical to Figure 1. Figure 3 shows the horizontal plan view of the model, the modeled wind directions (red arrow), and the LNG leak direction (blue arrow). The leak direction was chosen to be east because the leak would not be impinged and give a conservative result. The breach leaked horizontally at a height of 0.4 m above the ground.

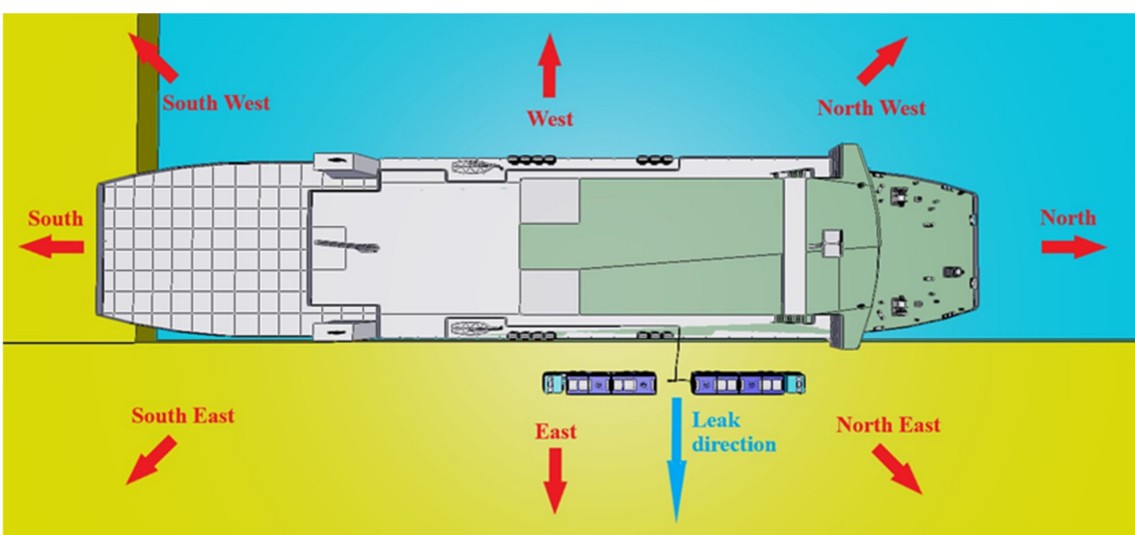

**Figure 3.** Plan view of the model, respective directions, and leak direction.

A structural non-uniform grid was applied by following FLACS best practice guidelines. The grid around the leak was locally refined to 0.257 m. The neighboring cells increased gradually at a rate of approximately 10% up to the domain boundaries. The ferry and trucks occupied a space of approximately 30 m by 115 m by 22 m. To allow the wind boundary to be established, the FLACS manual advises that the distance between object and domain be at least half the object dimension. Therefore, a grid domain covering 200 m by 300 m by 50 m in the x, y, and z-direction respectively was selected. The Ro-Pax ferry was placed approximately at the center of the simulated volume. Figure 4 shows the coarse grid setup.

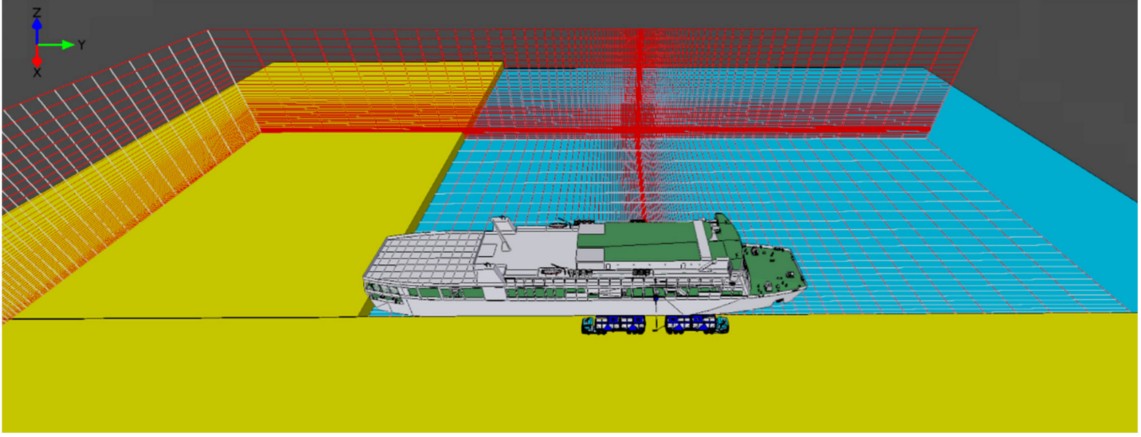

**Figure 4.** Coarse grid setup of the model.

A grid-independent study for four wind directions (south, west, north, and east) was conducted to test the coarse grid's suitability for the simulation. Table 1 shows the comparison of the coarse grid (406,980), 90% grid size (534,000), and 85% grid size (631,470) simulation result. The criterion for comparison is the furthest lower flammable limit (LFL) distance obtained from the different grid simulations. Please refer to Section 3 for a better explanation of LFL distance. The percentage difference ranges from 1.98 to 10.2%, demonstrating that the coarse grid size is adequate for the simulation.

**Table 1.** Comparison of coarse grid, 90% grid size, and 85% grid size simulation result.

| Wind Speed (m/s) | Direction | Mesh Type | LFL Distance (m) | Difference Percentage from Coarse Grid (%) |
|---|---|---|---|---|
| 8.1 (mean) | South | Coarse | 23.03 | N/A |
| | | 90% mesh size | 21.07 | −8.51 |
| | | 85% mesh size | 25.38 | 10.20 |
| | West | Coarse | 11.57 | N/A |
| | | 90% mesh size | 11.87 | 2.59 |
| | | 85% mesh size | 12.27 | 6.05 |
| | North | Coarse | 25.93 | N/A |
| | | 90% mesh size | 27.17 | 4.78 |
| | | 85% mesh size | 27.22 | 4.97 |
| | East | Coarse | 16.65 | N/A |
| | | 90% mesh size | 17.88 | 7.39 |
| | | 85% mesh size | 16.98 | 1.98 |

To measure the discretization error of the FLACS simulation, a grid refinement study based on Roache's work was performed [20]. The Roache grid convergence index (GCI) estimated the discretization error by comparing two discrete solutions from the coarse and refined grid. The GCI formula is calculated by:

$$\mathrm{GCI} = \frac{F_S}{\mathrm{r}^P - 1} \left| \frac{f_2 - f_1}{f_1} \right| \tag{9}$$

$F_S$ is the factor of safety set as 3, and r is the grid refinement factor set as 2 in this paper. P is the FLACS order of accuracy set as 2, $f_2$ is the coarse grid solution, and $f_1$ is the fine grid solution. The refined grid has 3,255,840 controlled volumes and is 8 times the coarse grid value (406,980). Figure 5 shows the refined grid setup.

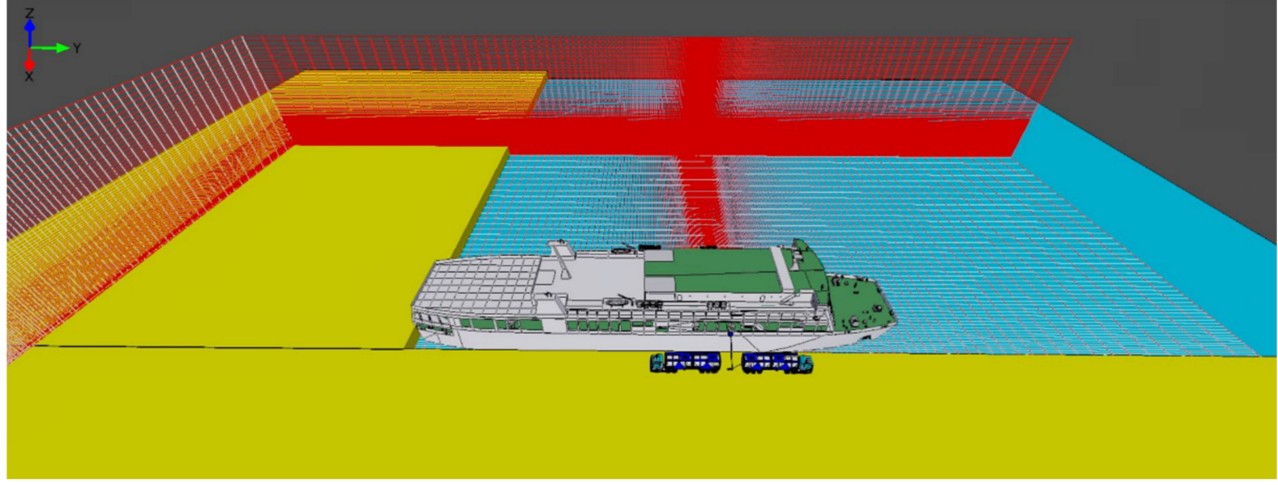

**Figure 5.** Refined grid setup of the model.

Based on Hong Kong January to May 2021 weather statistics, the initial conditions were set as an average temperature of 22.25 °C and 1 bar ambient pressure [21]. Ground roughness was set at 0.03 m because the terrain has few obstacles. The chosen solver was transient mode. Courant number (CFLC) based on sound velocity was set at 5. Courant number (CFLV) based on flow velocity was set at 0.5. The Hong Kong observatory wind record for January to May 2021 is presented as a wind rose in Figure 6. The prevailing wind direction is East North East (ENE). ENE mean wind speed is 8.1 m/s (elevation of 32 m above sea level) with a standard deviation of 2.38 m/s. Wind speed higher than the mean is associated with rainy weather which prevents LNG bunkering from taking place. Nubli and the team have concluded that lower wind speed leads to bigger gas dispersion [22]. To be conservative, only wind speeds at the mean speed and below were considered for the CFD simulation. Three wind speeds, 8.1 m/s (mean), 5.7 m/s (minus 1 standard deviation), and 3.3 m/s (minus 2 standard deviation) were selected for the simulation. These wind speeds cover the 5th to the 50th percentile wind speed distribution. The ENE wind profile was used for all the wind directions because no specific Hong Kong site was used in this study. Eight wind directions blowing north, northeast, east, southeast, south, southwest, west, and northwest were considered in the study. Pasquill stability class was set as neutral (D).

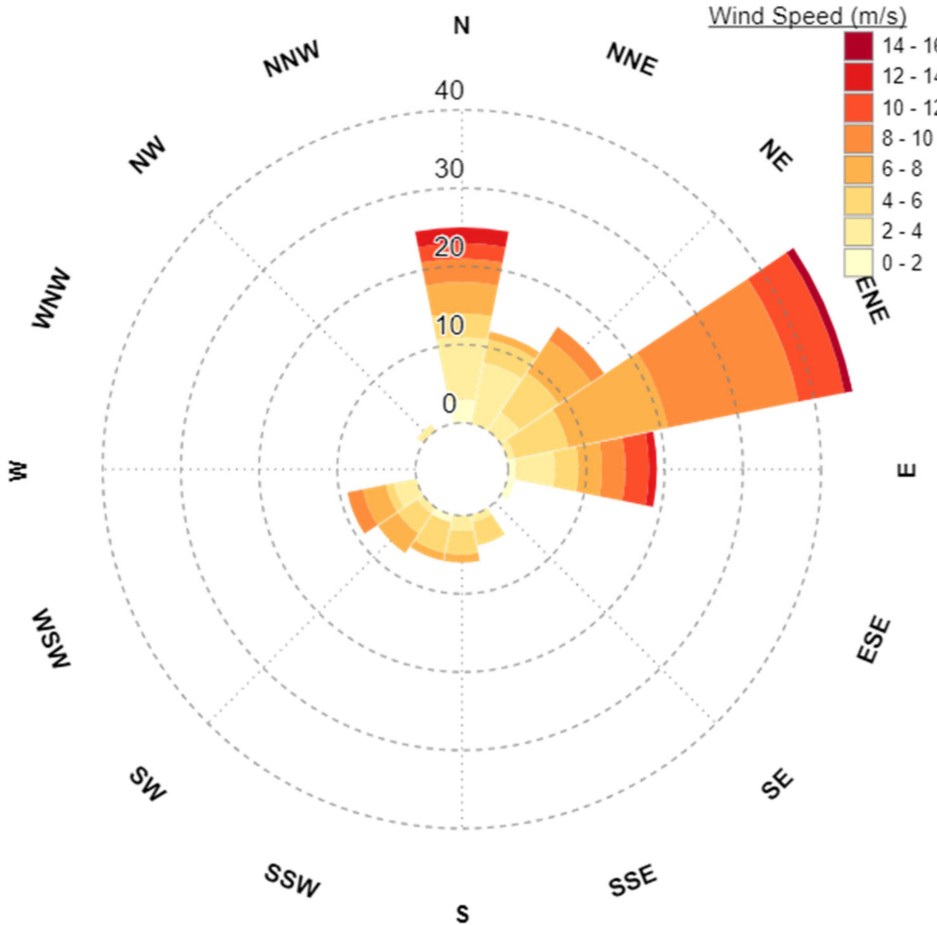

**Figure 6.** Windrose of Hong Kong weather from January to May 2021.

The initial leak condition was set as a 6 mm orifice releasing at 6 bar pressure. The leak duration was set as 120 s. An assumption was made that all the leaked LNG vaporized after leaving the containment and no LNG pool was formed. This assumption is supported by the recent leak experiment conducted by INERIS [23]. The experiment result showed that non-impinged pressurized LNG leaks from a breach less than 10 mm fully vaporize upon leaving containment. The Australian LNG source of 99.8% methane concentration

was set as the gas composition [24]. Given the extremely high methane concentration, LNG is considered pure methane in the simulation. Considering pure methane would give a conservative result because heavier alkane disperses at a slower rate.

## 3. Results and Discussion

### 3.1. FLACS Coarse Grid Results

Natural gas (NG) is flammable between 5% to 15% methane concentration in the air. The 5% concentration is called the lower flammable limit (LFL), and the 15% concentration is termed the upper flammable limit (UFL). The LFL distance covered by each coarse grid simulation is presented in Table 2. The LFL distance is the furthest distance from the leak source where the methane concentration has diluted to 5% in the air. All simulation was performed in a transient state and the numerical data was stored at a frequency of one second. The FLACS post-processing tool, Flowvis was used to visualize the LFL distance for the 120 s leaks. A total of 24 cases were simulated with FLACS and five cases (bolded in red) exceeded the safety zone (27 m) advised by the BASiL model. To be conservative, an assumption of 30% discretization error was applied to the results. All amplified results exceeding 27 m were repeated with the refined grid to reduce discretization error. Table 2 column "GCI study" shows 12 cases to be re-simulated with the refined grid.

**Table 2.** LFL distance of leaks from coarse grid simulation.

| Wind Speed (m/s) | Direction | LFL Distance (m) | 130% LFL Distance (m) | GCI Study |
|---|---|---|---|---|
| | South | 23.03 | 29.94 | Yes |
| | South-West | 17.05 | 22.17 | No |
| | West | 11.57 | 15.04 | No |
| 8.1 (mean) | North-West | 20.32 | 26.42 | No |
| | North | 25.93 | 33.71 | Yes |
| | North-East | 25.17 | 32.72 | Yes |
| | East | 16.65 | 21.65 | No |
| | South-East | 11.28 | 14.66 | No |
| | South | 26.83 | 34.88 | Yes |
| | South-West | 21.71 | 28.22 | Yes |
| | West | 20.65 | 26.85 | No |
| 5.7 (1 standard deviation) | North-West | 17.47 | 22.71 | No |
| | North | 21.07 | 27.39 | Yes |
| | North-East | 28.42 | 36.95 | Yes |
| | East | 20.65 | 26.85 | No |
| | South-East | 14.28 | 18.56 | No |
| | South | 31.73 | 41.25 | Yes |
| | South-West | 17.81 | 23.15 | No |
| | West | 29.95 | 38.94 | Yes |
| 3.3 (2 standard deviation) | North-West | 20.27 | 26.35 | No |
| | North | 29.37 | 38.18 | Yes |
| | North-East | 27.92 | 36.30 | Yes |
| | East | 26.15 | 34.00 | Yes |
| | South-East | 20.73 | 26.95 | No |

### 3.2. FLACS Refined Grid Results

Table 3 provides the GCI value, LFL distance of the refined grid results, LFL distance range, and BASiL model percentile ranking. Using Equation (1), the GCI value was calculated from the coarse and refined grid LFL distance. The GCI value estimates the discretization error percentage to quantify the upper and lower LFL uncertainty boundaries. For example, a GCI value of 0.09 means the true LFL distance is between 91% and 109%

of the refined grid LFL distance. The percentile ranking gauges the accuracy of the BASiL model and is calculated as follows:

$$\text{Percentile} = 100 \times \frac{27 - \text{lower LFL distance}}{\text{Upper LFL distance} - \text{lower LFL distance}} \tag{10}$$

**Table 3.** LFL distance of leaks from refined grid simulation.

| Wind Speed (m/s) | Direction | GCI Value | LFL Distance (m) | LFL Distance Range (m) | BASiL Model Percentile Ranking |
|---|---|---|---|---|---|
| 8.1 (mean) | South | 0.098 | 25.53 | 23.03–28.03 | 79.40 |
| | North | 0.017 | 26.37 | 25.93–26.81 | 121.59 |
| | North-East | 0.090 | 27.67 | 25.17–30.17 | 36.60 |
| 5.7 (1 standard deviation) | South | 0.063 | 28.63 | 26.83–30.43 | 4.72 |
| | South-West | 0.126 | 19.28 | 16.85–21.71 | 208.85 |
| | North | 0.225 | 27.17 | 21.07–33.27 | 48.61 |
| | North-East | 0.009 | 28.17 | 27.92–28.42 | N/A |
| 3.3 (2 standard deviation) | South | 0.114 | 35.83 | 31.73–39.93 | N/A |
| | West | 0.191 | 25.15 | 20.35–29.95 | 69.3 |
| | North | 0.240 | 38.62 | 29.37–47.87 | N/A |
| | North-East | 0.282 | 21.77 | 15.62–27.92 | 92.52 |
| | East | 0.048 | 24.95 | 23.75–26.15 | 135.42 |

If the lower LFL distance boundary is more than 27 m, no percentile is given because the BASiL model value does not fall within the boundaries. The highest GCI value is 0.282, roughly 2% less than the assumed discretization error for using the coarse grid. Hence, in cases that were not checked with the refined grid, it is certain that the LFL distance did not exceed 27 m.

Using 50 percentile ranking as the passing criteria, there are three cases in which the BASiL model may have underpredicted the safety zone which are bolded red in Table 3. There are three cases in which the BASiL model underpredicted which are listed as N/A in the percentile ranking column. Figure 7 shows the gas dispersion profiles (minimum 5% gas concentration in air) of the underpredicted cases, 5.7 m/s northeast wind, 3.3 m/s south wind, and 3.3 m/s north wind sequentially. The blue cloud shows the location where the methane concentration is at least LFL level. The paths where the dispersion exceeded 27 m in radius are generally in the wind direction except for the northeast wind case. Figure 8 shows the wind streamlines on the XY plane at 1 m above the ground. The different color depicts the wind speed in different regions. The leak area is blocked from the incoming wind and causes the nearby region air to flow toward the manifold. The black lines outline the streamlined formation. The leaked plume disperses in the streamlined direction and the result is shown in Figure 7a.

None of the leaks disperse beyond 3 m height and this observation is explained by the stratification effect of a cryogenic gas leak. Methane gas has negative buoyancy relative to air at a temperature below −117 degrees Celsius [5]. Therefore, the flammable gas spreads more in the horizontal axes than in the vertical direction.

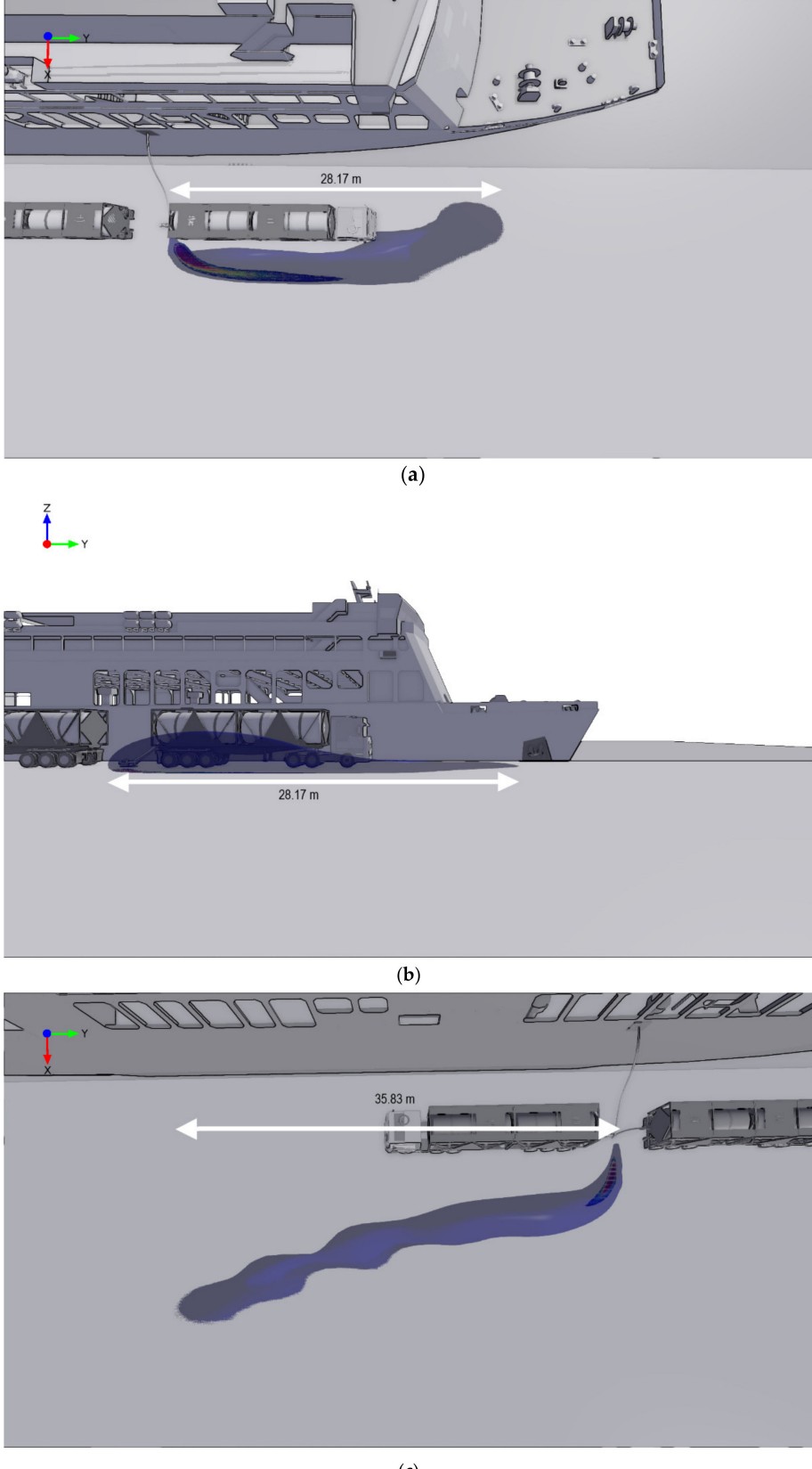

(a)

(b)

(c)

**Figure 7.** *Cont.*

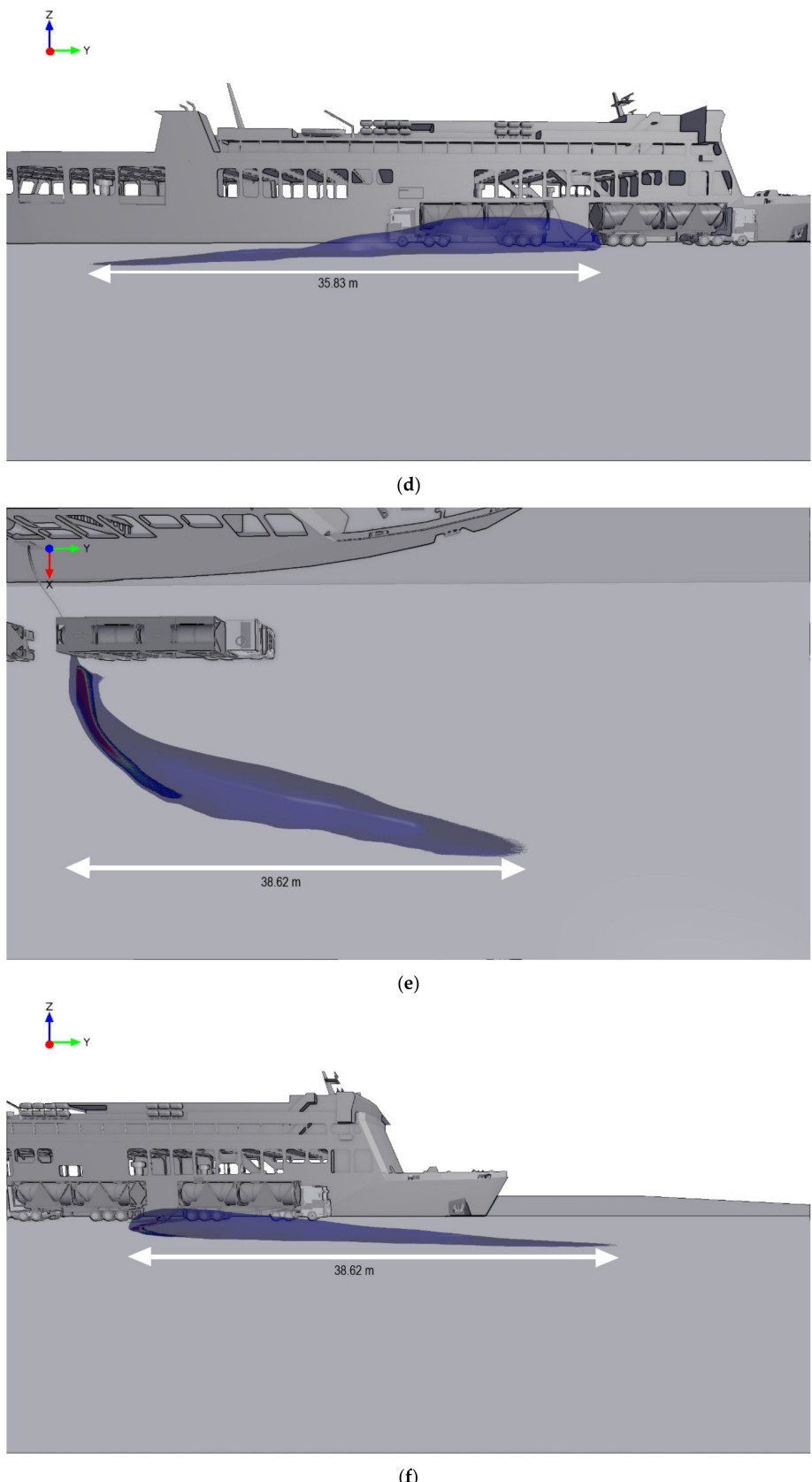

**Figure 7.** Snapshots of LNG dispersion profiles that exceeded 27 m: (**a**) Plan view of 5.7 m/s northeast wind, (**b**) Section view of 5.7 m/s northeast wind (**c**) Plan view of 3.3 m/s south wind, (**d**) Section view of 3.3 m/s south wind, (**e**) Plan view of 3.3 m/s north wind, (**f**) Section view of 3.3 m/s north wind.

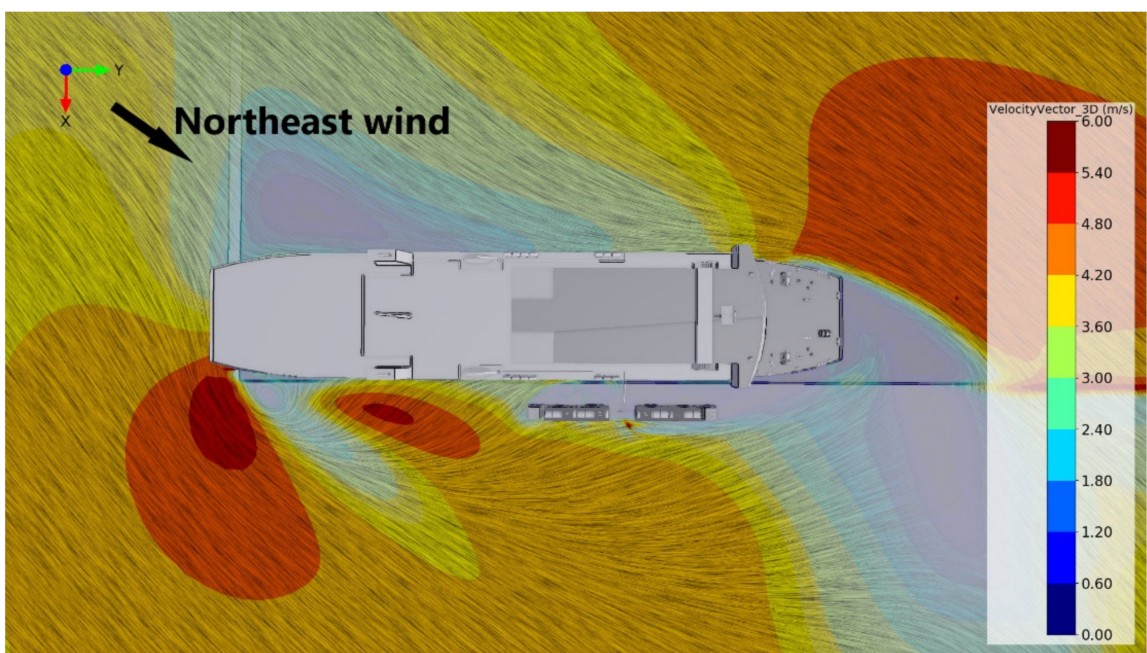

**Figure 8.** Wind streamlines with a 5.7 m/s northeast wind.

Table 4 summarizes the validation of the FLACS result with the BASiL model. Of 24 FLACS simulations, 18 cases did not exceed the 27 m safety zone advised by the BASiL model. Three cases were inconclusive as the BASiL model value falls in the lower 50th percentile of the LFL uncertainty boundaries. From the GCI analysis, three cases of gas dispersion were certain to have exceeded the 27 m. The cases in which BASiL had underestimated the safety zone are for wind speeds below the mean. This is unsurprising as BASiL is based on data interpolation to advise the safety zone. Assuming the 1.4 million-combination database has more information on parameter values centering near the normal values, it may be unsuitable to capture the 3.3 m/s wind speed cases which are outliers (5% percentile) in the wind distribution curve. The average LFL distance of the 12 cases in Table 4 is 27.4 m which is close to the 27 m advised by BASiL. This finding demonstrates the interpolation methodology nature adopted by BASiL.

**Table 4.** Summary of FLACS results compared with BASiL model.

| Wind Speed (m/s) | Direction | BASiL Model Safety Zone Exceeded |
|---|---|---|
| 8.1 (mean) | South | No |
| | South-West | No |
| | West | No |
| | North-West | No |
| | North | No |
| | North-East | No |
| | East | No |
| | South-East | Uncertain |
| 5.7 (1 standard deviation) | South | Uncertain |
| | South-West | No |
| | West | No |
| | North-West | No |
| | North | Uncertain |
| | North-East | Yes |
| | East | No |
| | South-East | No |

**Table 4.** *Cont.*

| Wind Speed (m/s) | Direction | BASiL Model Safety Zone Exceeded |
|---|---|---|
| | South | Yes |
| | South-West | No |
| | West | No |
| 3.3 | North-West | No |
| (2 standard deviation) | North | Yes |
| | North-East | No |
| | East | No |
| | South-East | No |

## 4. Conclusions

FLACS was used to simulate 24 LNG horizontal leaks covering wind speed distribution up to the 5th percentile for a Ro-Pax ferry bunkering. GCI analysis was performed to reduce the CFD discretization error. The BASiL model safety zone estimation did not exceed FLACS findings except when wind speed was below the mean. Three validation cases have inconclusive findings and need further investigation. Future work would be to perform these three cases with four times grid refinement based on this paper's coarse grid setup. With three discrete solutions using a constant grid refinement of 2, the GCI safety factor can be reduced from 3 to 1.25 [20]. This would reduce the discretization error uncertainty and give a conclusive answer as to whether the BASiL model has underestimated the safety zone. However, this refinement will have approximately 26 million controlled volumes and require significant computational resources. The average computing time to perform the coarse grid simulation (406,980 controlled volumes) is 12.5 h and for the fine grid simulation (3,255,840 controlled volumes) is 200 h. It will need at least one month of computing time to perform simulations for 26 million controlled volumes with a commercial computer.

Other future work would be to examine ways to reduce the LFL distance for the leak cases, especially for wind speed conditions at and below the 5% percentile distribution. One potential solution is putting up a barrier to redirect the gas cloud. As observed in the northeast wind simulation, wind blockage can redirect the gas plume. This phenomenon may have the potential to aid in gas mixing and significantly influence LNG dispersion. The approach to studying port arrangement impact on LNG dispersion can be adopted from similar work done by Mariotti and team [25]. By reducing the LFL distance for the worst dispersion case, the ship owner can use the free space for simultaneous operation, thereby reducing the time spent in port.

**Author Contributions:** Conceptualization: B.H.L.; Methodology: B.H.L.; Formal analysis and investigation: B.H.L.; Writing—original draft preparation: B.H.L.; Writing—review and editing: B.H.L. and E.Y.K.N.; Funding acquisition: B.H.L. and E.Y.K.N.; Resources: B.H.L. and E.Y.K.N.; Supervision: E.Y.K.N. All authors have read and agreed to the published version of the manuscript.

**Funding:** This work was supported by the Industrial Postgraduate Programme (#001799-00001) initiated by the Singapore Economic Development Board (EDB).

**Institutional Review Board Statement:** Not applicable.

**Informed Consent Statement:** Not applicable.

**Data Availability Statement:** The data presented in this study are available on request from the corresponding author. The data are not publicly available due to the academic license agreement.

**Acknowledgments:** The authors would like to express their appreciation to Gexcon Bjørn Lilleberg for his advise on using FLACS.

**Conflicts of Interest:** The authors declare no conflict of interest.

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
