# Peer review of "Comparison of FLACS and BASiL Model for Ro-Pax Ferry LNG Bunkering Leak Analysis"

_fluids, doi:10.3390/fluids7080272_

Round 1

Reviewer 1 Report

The authors carried out a CFD case study of LNG bunkering leak to evaluate the accuracy of prediction for safety zone by BASiL model. The commercial code FLCAS was used to simulate the LNG leak at a RoPax ferry bunkering. Different wind speeds and directions were used. The manuscript provides useful information for evaluation of BASiL model. it is generally written well. I recommend publication after mandatory revisions. My specific comments are

(1) the authors claimed they performed grid refinement study systematically to quantify the discretization error uncertainty. I can not agree with it. Typically, a number of grid refinement levels need to be used to be sure that the numerical results are grid independence. The authors only used two grid levels (one is coarse, the other one is fine mesh) in their grid-independent study. It is far from enough. I recommend the authors use more grid levels to make sure numerical results are grid independent.

(2) The authors need to justify why using Eq. 7 to measure the discretization error, as it is not widely used in CFD area.

(3)It is better to include the computation time for each case.

(4)Is wind boundary condition only applied at one face of the computational domain? More details are needed for boundary conditions.

(5)The authors mentioned the computational domain is 200mX300mX50m. How did they determine the size of the domain? How did they know the size was large enough so the solutions were not size dependent?

(6)The governing equations presented are transient PDEs, so the solutions are time dependent. Did the authors solve the equations until the steady state? Or they the select a time to report the results (e.g., predicted safety zone size, figs. 7,8)? More information is needed regarding this. 

Reviewer 2 Report

The authors compare the safety zone estimated by Bunkering Area Safety Information for LNG (BASiL) model and computational fluid dynamic (CFD) software FLACS for a Ro-Pax ferry bunkering. The aim of the paper is clear and interesting to me. The paper can be considered for publication after the authors have replied to the following mandatory remarks.

1) Some details of the CFD simulation setup are not given. Please provide the p-v coupling, the order of accuracy of the discretization schemes, and the convergence criterion. 

2) As for the CFD simulations, the description of the vortical structures should be given. These structures, when present, may significantly influence the LNG dispersion. Among the different vortex indicators, I suggest the vortex indicator lambda2 (Jeong and Hussain, 1995). This criterion has been already used for the comparison between different configurations (e.g., Mariotti et al., 2019). The authors should at least mention the criterion, and show some numerical results.  

Suggested references: 

- J. Jeong, F. Hussain, On the identification of a vortex, J. Fluid Mech. 285, 69–94 (1995)

- A. Mariotti, C. Galletti, E. Brunazzi, and M.V. Salvetti, “Steady flow regimes and mixing performance in arrow-shaped micro-mixers,” Physical Review Fluids 4, 034201 (2019b)

Round 2

Reviewer 1 Report

I'm satisfied with the revised manuscript. I recommend publication as is.

Reviewer 2 Report

accept in present form